# Navigating challenges: a socioecological analysis of sexual and reproductive health barriers among Eritrean refugee women in Ethiopia, using a key informant approach

Nejimu Biza Zepro ![ORCID] ,[1,2,3] Araya Abrha Medhanyie,[4] Nicole Probst-Hensch,[1,2] Afona Chernet,[1,2] Rea Tschopp,[1,2] Charles Abongomera,[1,2] Daniel H Paris,[1,2] Sonja Merten ![ORCID] [1,2]

¹Department of Epidemiology and Public Health, Swiss Tropical and Public Health Institute, Basel, Switzerland
²University of Basel, Basel, Switzerland
³College of Health Sciences, Samara University, Afar, Ethiopia
⁴School of Public Health, College of Health Sciences, Mekelle University, Tigray, Ethiopia

**Correspondence to**
Nejimu Biza Zepro;
nejimu.zepro@swisstph.ch

## ABSTRACT

**Objectives** The study aimed to explore the experiences and perceptions of healthcare providers (HCPs) regarding the sexual and reproductive health (SRH) challenges of Eritrean refugee women in Ethiopia.

**Design** A qualitative exploratory design with the key informant approach.

**Setting and participants** The study was conducted in the Afar regional state, North East, Ethiopia. The study participants were HCP responsible for providing SRH care for refugee women.

**Results** Eritrean refugee women have worse health outcomes than the host population. The SRH needs were found to be hindered at multiple layers of socioecological model (SEM). High turnover and shortage of HCP, restrictive laws, language issues, cultural inconsistencies and gender inequalities were among the main barriers reported. Complex multistructural factors are needed to improve SRH needs of Eritrean refugee women.

**Conclusions** A complex set of issues spanning individual needs, social norms, community resources, healthcare limitations and structural mismatches create significant barriers to fulfilling the SRH needs of Eritrean refugee women in Ethiopia. Factors like limited awareness, cultural taboos, lack of safe spaces, inadequate healthcare facilities and restrictive policies all contribute to the severe limitations on SRH services available in refugee settings. The overlap in findings underscores the importance of developing multilevel interventions that are culturally sensitive to the needs of refugee women across all SEM levels. A bilateral collaboration between Refugees and Returnees Service (RRS) structures and the Asayta district healthcare system is critically important.

## STRENGTHS AND LIMITATIONS OF THIS STUDY

⇒ Key informant researcher approach is found to be appropriate to explore super sensitive sexual and reproductive health (SRH) issues such as sexuality, multiple sex partners and intimate partner violence.
⇒ Qualitative study design allowed for a more detailed and nuanced contextual understanding of the SRH challenges among refugee women in such fragile hard-to-reach settings.
⇒ Collaboration with an experienced social science researcher and local institutions was instrumental in building relationships and trust between the research team.
⇒ Comparing results with other sites may be difficult considering that the study was conducted with a small sample of healthcare providers in a defined humanitarian setting.

and well-being for all.[1] However, the public health priorities during migration on sexual and reproductive health (SRH) needs are often overlooked by the health system, with staggering consequences.[2] WHO advocates for universal health coverage, leaving no one behind. This includes the most vulnerable populations including refugees.[1] WHO has also recognised SRH care access as a global health priority in every setting.[3]

Previous studies have shown that SRH service needs of refugee women are often prone to poor access to SRH key services, risky life-threatening complications, traumatic events to serious mental illness, unmet need to contraceptives, unplanned pregnancy, unsafe abortion, sexual gender-based violence and sexually transmitted infections (STIs).[4–6] Patriarchy and violence also impacted SRH of Rohingya refugee women in Bangladesh.[5 7] The United Nations High

## INTRODUCTION

Migration health is a growing global public health issue that should be addressed in line with the United Nations (UN) 2030 Sustainable Development Goals (SDG), particularly Goal 10, reducing inequalities; goal 5, gender equity; and Goal 3, good health

Commissioner for Refugees (UNHCR) estimates that forced migration has increased from 42.8 million in 2012 to 101.1 million in 2022 as a result of persecution, hostilities, conflict, violence, abuses of human rights and significant disruptions to the public order.[8]

Ethiopia is the second most populous country in Africa (over 110 million people) and the third-largest refugee-hosting African country, sheltering 926 000 refugees and asylum seekers as of 13 June 2023, mainly from South Sudan, Somalia and Eritrea.[9–11] In addition, a total of 2.8 million internally displaced persons (IDPs), excluding IDPs from the Tigray region, have been identified.[12] Over 140 000 Eritreans who fled hardship and persecution in their home country are hosted in Ethiopia.[13] Some 96 223 Eritrean refugees lived in four refugee camps in the northern part of Ethiopia, Tigray region.[14] However, the 2-year civil war that erupted in the region in November 2020 affected the area including these camps and triggered a massive exodus of local civilians and Eritrean refugees to Sudan and internally to a nearby Afar region, which created a huge humanitarian crisis. The Afar region hosts nearly 300 000 IDPs due to the war in Tigray. IDPs have settled in the collective sites with host communities. More than 4000 IDPs live in Semera, and another 10 000 live in Afdera town.[15] This all creates a major burden on an already overburdened country.[16]

Migration negatively impacts everyone; particularly, women, girls and children are the most vulnerable groups.[17] Nearly half, 48.4% and 46%, of the world's migrant population are women and children, respectively.[8 18] Refugee women face a double burden; in addition to the hardships of forced migration, they suffer from challenges related to their gender. They are at increased risk for sexual abuse, rape, unwanted pregnancies, unsafe abortions and the spread of STIs, including HIV, and short birth interval, which lead to long-term mental health problems and high infant and maternal mortality.[19–22] During the collapse of existing social systems, women are unable to care for their children and themselves. In host countries, migrant women usually bear full responsibility for family members and child care but at the same time do not have access to resources and the ability to make autonomous decisions about them to the same extent as their male counterparts.[22–25]

For this study, the definition of SRH was adopted from the International Conference on Population and Development (ICPD 1994) and the United Nations Population Fund, as it refers to a state of complete physical, mental and social well-being in all matters relating to the reproductive system. It implies the ability to be free from unwanted pregnancy, unsafe abortion, STI including HIV, prevention of violence against women, elimination of harmful traditional practices and all forms of sexual violence.[26] SRH is at the heart of human existence, prosperity and well-being of individuals and societies worldwide.[27] SRH is a core component of primary healthcare, a universal human right, which applies to every humankind including refugees, internally displaced people and

asylum seekers but is often forgotten.[7 16] To realise this universal right, all affected groups must have access to SRH services so they are free to make informed choices and decisions.[1 16] SRH is a basic human right and an essential element in achieving health, gender equity and social justice. Women's rights to SRH were recognised as fundamental human rights at the ICPD 1994 in Cairo and went beyond refugee camps and other humanitarian emergencies.[28–30] Refugee women continue to experience poorer SRH outcomes than the general population, and refugee women are over-represented in the numbers of poor SRH outcomes.[30–32]

The socioecological model (SEM) is a theoretical framework that suggests individual behaviour, and health outcomes are influenced by a combination of individual, interpersonal, social, organisational and environmentally complex inter-related factors. This model emphasises one interconnectedness of these factors and the need to consider them holistically by addressing public health issues. Bronfenbrenner first introduced the SEM model as a conceptual framework for comprehending human growth.[33] In the 1980s, the model was organised as a theory.[33] The modified SEM in this study seeks to comprehensively assess individual, community, organisational and societal levels.[34] These five levels of SEM are highly interconnected.[35] Healthcare providers (HCPs) themselves experience challenges at all five levels of the SEM framework. Women's past experiences with contraceptives may present challenges to using SRH services at the individual level.[36 37] On the interpersonal level, the lack of cultural competence of healthcare staff and communication difficulties must be considered.[36] Organisational-level challenges were women's unfamiliarity with the healthcare system, a lack of training and protocols governing the delivery of healthcare.

Although Ethiopia's healthcare system is still developing, SRH care is provided free of charge as part of the public service at the primary care unit. Services are mainly provided in the Maternal and Child Health Unit in health centres, health posts and district hospitals. Pregnancies with complications or at risk of complications are treated in hospitals as part of specialised healthcare. Mainly, midwives and nurses provide their services independently in health centres.

SRH challenges are recognised from the end-user (refugee) perspective; there is a lack of research on HCP perceptions and experiences of interacting with refugee women seeking SRH care. The perspectives of HCP are equally important, as they often deal with, provide care to and support refugees. HCPs are key informant researchers (KIRs) as they are in a better position to tell what challenges and opportunities the health system poses to the refugees.[38] Moreover, since refugees cannot tell or inform the barriers and opportunities that they may have from the health system, they also play a role in helping refugees navigate the healthcare system and access the services they need.[35] However, only few researchers have looked at SRH needs of refugee women from HCP's

perspectives.[36 37 39 40] Thus, we want to uncover how different broad-level ecological factors impact women's SRH. Therefore, a systematic approach that considers influences that go beyond the individual level is relevant to gain a broader understanding of health service utilisation.[35] This study examined the difficulties of HCP when caring for refugee women and examined how these difficulties affected the provision of SRH care.

Findings from the study could help the RRS agency to improve SRH service provision for refugees and migrants. Likewise, other major stakeholders like the UN, UNHCR, World Food Programme, International Medical Corps Goal Ethiopia and others will benefit from the study findings. This will also allow for exploring the possibility of updating and standardising service and training packages for SRH service providers to improve comprehensive SRH service provisions in similar humanitarian settings. Thus, understanding the magnitude of the problems and the reasons behind them is crucial for programme planners to design effective preventive strategies.

## METHODS
### Study setting
The study was the qualitative arm of the Novel Integrated Infectious Disease Surveillance (NIIDS) project carried out in Ethiopia and Switzerland. The study was conducted in the Asayta refugee camp in Afar, North East part of Ethiopia, which currently hosts more than 18 000 Eritrean refugees. The district is characterised by an arid climate and an ecologically fragile ecosystem, making it difficult to provide basic healthcare and life-saving services. It is located at 65 km South West of Semera (administrative capital) and 649 km from Addis Ababa, the capital city of Ethiopia, and shares an international border with Djibouti. The temperature varies between 30°C and 45°C, the altitude ranges between 350 and 500 m above sea level and the rainfall is irregular. The total population of the district is estimated at 47 210 where 66% live in rural areas. They rarely use healthcare facilities, mainly because they do not know where to get healthcare and access to services.[41] There are four private clinics, three public health posts and one public district hospital. Malaria, diarrhoea and other febrile illnesses are quite common in the district due to inadequate clean water and sewage disposal facilities, inadequate availability of health facilities, low supplies and delays in the distribution of humanitarian relief supplies.[41]

Asayta refugee camp, which is located 2 km outside Asayta town, is administered by the RRS in collaboration with UNHCR. The refugee clinic is located inside the camp that provides primary healthcare to Eritrean refugees. Severe illnesses that require advanced diagnostics are referred to the nearby Asayta district hospital.

### Study design
An exploratory qualitative study design with a KIR approach was applied.[42] Semistructured individual interviews were conducted to explore the experiences and perceptions of HCP in providing SRH care for Eritrean refugee women.

### Study participants
Study participants were HCPs responsible for providing SRH care to Eritrean refugee women in the Asayta refugee camp. HCPs were purposely selected because they are particularly aware of SRH needs and service uptake among Eritrean refugee women. The perspectives of HCP are equally important in research. By sharing their knowledge and experience, HCP can help ensure that research studies are relevant, culturally appropriate and feasible in the context of the refugee women being studied.

### Patient and public involvement
No patients were involved. Key informants (KIs) were not involved in the design, conduct or analysis of this paper. However, we are planning to hold a dissemination workshop on the project's key findings and consider future follow-up projects which will result in a long-term partnership with the Asayta refugee camp.

### Data collection tool and participant recruitment
We developed a separate semistructured initial qualitative interview guide based on a review of the relevant literature.[43–46] Study participants were requested to mention their professional background, experience, health facility responsibilities and the proportion or number of refugee women they serve each month. Participants were then asked a series of open-ended questions about their experience in providing SRH care for Eritrean refugees.

Data were collected through Key Informant Interviews (KIIs) and focused observations. The initial interview guide was developed to comprehensively capture a core set of questions from all participants. Further questions were posed depending on the type of occupation and organisation in which the interviewees worked. The KII guide included general questions about the SRH needs of women as perceived by HCP during the delivery of SRH services, the accessibility of SRH services, the challenges women and HCP face during treatment, and recommendations on how to mitigate these challenges (online supplemental file 2). To increase the credibility and richness of the data, respondents were intentionally selected to be as diverse as possible in terms of work experience, gender, work location and cultural background. Field notes were taken to supplement the audio recordings and document non-verbal cues. At the end of all sessions, the lead author debriefed the participants, summarising basic information.

The first author conducted the KII with the support of a trained research assistant who had no prior contacts or relationships with study participants. Before the interviews began, a researcher explained the detailed objectives of the study to the participants and obtained their written and verbal consent to conduct the interview. Study

participants signed a written informed consent form, read the confidentiality statement and were reminded of their voluntary participation and their right to terminate the interview at any time. The data collection period was between July and August 2022.

Interviews were conducted face to face in the Amharic language (Ethiopia's official language). Interviews were recorded and transcribed verbatim. Data saturation was reached when no more new information continued to emerge. A summary report of each interview was also written. After the transcriptions were finished, the audio files were removed from each transcriber's personal computer. The interviews were professionally transcribed by the first author with the assistance of an experienced female qualitative researcher. The transcriptions were translated into English for further analysis. On the study materials or transcripts, there were no names or other identifiable personal information. COVID-19 protection measures were taken during interviews, including the use of hand sanitisers, the use of face masks, social distancing and conducting interviews in ventilated large rooms. To maintain anonymity and confidentiality, all study participants were identified with a code.

### Data analysis

The audio recordings were listened to several times to confirm the completeness of the verbatim transcription, identify contextual meanings and make preliminary interpretations of the data. This yielded sufficient textual data to perform iterative categorisation of the data for better analysis. The coding process helped us to identify common themes. Codes with similar meanings were combined and then sorted to form subcategories. The subcategories were then compared, and categories with similar concepts were combined into one category.

Due to the complexity of SRH topics, especially in such groups of women, we conducted a systematic two-stage analysis. First, we conducted a thematic analysis of the interviews. Second, taking into account the broader nature of the research question, we developed an adapted SEM model to provide a clear framework and recommendations for improving the SRH needs of Eritrean refugee women. We opted for this model because it effectively illustrates the complex interplay among various factors influencing the health and well-being of refugees.[22 47] These factors encompass individual aspects such as behaviours, social and community elements like norms, institutional and policy components such as the healthcare system, as well as structural factors including policy and protection. The SEM model has been used elsewhere to understand SRH barriers for refugees. For example, Shtarkshall *et al* and Mengesha *et al* applied this model to examine refugee and migrant healthcare access issues in Israel and Australia.[39 48] Mengesha and colleagues, Keygnaert and colleagues, Darebo and colleagues, and Wells and colleagues have used the SEM framework as well.[22 39 49 50] This model proved to be particularly relevant because it focuses on the synergistic relationship between

individuals and their social environment.[33 39] The model also describes the multiple interactions among individual (micro), interpersonal (meso), institutional (exo) and societal (macro) factors that influence health seeking behaviours.[22 39 47]

Computer-assisted qualitative and mixed-method data analysis software (MAXQDA 2018) was used to organise codes and themes to support analysis through close reading of transcripts and notes. Common themes were discussed further by the authors for relevance and consistency with the SEM framework. The use of SEM is justified by the fact that refugee women's use of SRH care is influenced by multiple dynamic factors that consider the interactions between different levels of SEM.[22 51 52] Again, it is the interactions of refugee women with their environment, at individual, interpersonal, community, societal, institutional and policy levels, that can influence access and utilisation of SRH care by refugee women, as well as SRH care provision by HCP in Ethiopia.[51] The data transcripts were then analysed using the Gale *et al* framework method, drawing from Braun and Clarke's thematic analysis.[53 54] The analysis steps suggested by Gale *et al* were modified and applied as follows: (1) audio transcription and translation, (2) reading and rereading the translations to become familiar with the interviews, (3) line-by-line coding using MAXQDA, (4) developing a thematic framework, (5) applying the framework, (6) placing the data into a SEM framework and (7) interpreting the data.[54]

Inductive approach was used to identify codes and themes from the data. Analysis began with a familiarisation to the developed transcripts to assign initial codes. The data coding was done by using MAXQDA, and integrity was double-checked. A summary of the coded data set was then created with specific reports to identify initial themes. This process allowed to define core concepts and identify unique and specific stories for each theme. Finally, SEM was applied to identify and interpret factors that influence SRH care among Eritrean refugee women at the individual, interpersonal, institutional, societal and policy levels.

### Data quality

We took part in a reflective process relevant to the dynamics between the researcher and the researched by examining critically how our social and cultural backgrounds, assumptions and positions impacted the study process.[55] Researcher reflexivity was ensured through additional field notes, focused observations, informal conversations with concerned stakeholders and peer debriefings after each interview. Data dissemination and reporting were guided by the Consolidated Criteria for Reporting Qualitative Research and the Standards for Reporting Qualitative Research guidelines (online supplemental file 1).[56 57] Rigour and trustworthiness were assessed using the principles of credibility, reliability, confirmability and transferability. Data credibility was enhanced through prolonged engagement, peer

debriefing, written reminders and various bracketing methods for data collection. A complete and continuous data collection method was used to ensure data reliability. The data were also reviewed by an expert researcher who had no connection to the study and was an outside observer. To confirm the data, the entire research process was recorded. Also, research colleagues who were not involved in the study but were knowledgeable about qualitative data analysis were given access to the transcripts of interviews as well as the extracted codes and categories to confirm the validity of the coding procedure. Finally, to ensure the transferability of the data, results were examined from several individuals who had similar characteristics to the participants in the study but did not participate in the present research. Data triangulation was achieved through a combination of different data sources (interviews, field notes, focused observation and SEM theory).

The research team's interdisciplinary makeup allowed for a high level of reflexivity, which pushed us to investigate and consider assumptions and biases. All the authors have ample experience with qualitative studies, and two of the authors (SM and AAM) are recognised experts in the field of SRH in fragile humanitarian settings. We looked at how closely the retrieved themes and theme clusters matched the SEM categories. Almost 72% of the codes and themes developed were in agreement with those taken from the SEM perspective and applied by other researchers[35 39 48]

## RESULTS

10 KIIs took part in the study, with 6 from the refugee camp and 4 from the host hospital. Among them were five midwives, four nurses and one senior manager. The duration of the interviews ranged from 30 to 72 min. The results of the study are presented thematically. The various codes that emerged were classified into several categories and subcategories. The overlapping and similar categories and subcategories were grouped into five main themes. Finally, we found that many themes overlapped at multiple levels of the SEM framework, indicating multi-layered influences of complex issues.

### Sociodemographic characteristics of the respondents

The total of 10 HCPs participated in the study with a gender balance of 5 men and 5 women. The age range of participants was between 20 and 40 years, indicating that most of the study participants were young employees. The minimum education level was a college diploma, while more than half of the participants had a bachelor's degree. All participants except two were not from Afar region. Most of them had a total of more than 3 years of work experience, while the rest had been on the job for only 2 or 3 years. The experience of working in Asayta refugee camp is quite limited with a maximum of 1 year. We cannot list the study participants here, as they are few and easily identified.

### Themes and categories according to SEM

The analysis of the main findings of the present study relating to the SRH needs of Eritrean refugee women was explored by thematically mapping the problems to five SEM levels: individual, interpersonal, organisational, community and larger policy perspectives. The intersecting rings in the model show how factors at one level influence factors at other levels (figure 1).

### Individual-level/intrapersonal-level factors

The first SEM level, individual level, considers personal characteristics such as age, gender, knowledge, culture, beliefs and behavioural and lifestyle factors. Job category, camp experience, gender, ethnicity, religion, marital status, SRH knowledge, displacement experiences and cultural taboos were identified as individual factors that impede appropriate SRH service delivery. However, Eritrean refugee women were strongly influenced by their SRH knowledge, autonomy, cultural beliefs, challenges from HCP and stressors of adapting to a new environment.

### Refugee women's SRH knowledge

A lack of clear and accessible health information is mentioned as a barrier to accessing SRH care. HCP noted that SRH services are not adequately advertised in local languages as well as refugee's mother tongue, leading to confusion among women, particularly Eritrean refugee women who have limited literacy skills or are unfamiliar with the health system.

It may be helpful to consider strategies to increase the accessibility and visibility of SRH services in the refugee camp, such as using multiple communication methods (eg, visual art and outreach), providing information in local languages and clearly labelling the location of services.

It was explained that the refugee camp seeks to provide health education on a range of SRH topics, including STI, contraceptive methods, prenatal care, obstetric care, postpartum care and gender-based violence. Women must have access to accurate and relevant information about SRH issues so they can make informed decisions about their health.

However, women generally have a limited understanding of SRH issues. Women turn to multiple people for information on SRH issues, and HCPs report having to spend a great deal of time explaining and persuading women. This could be due to several factors, including cultural beliefs and practices, lack of access to education and lack of exposure to SRH information in addition to language barriers.

> Eritrean refugee women's SRH knowledge is quite limited. We spend a lot of time educating refugee women about SRH issues because this information is not a high priority as there are more pressing issues for refugee women, such as housing, food, asylum applications, and social issues. They want clear and complete information about prenatal care,

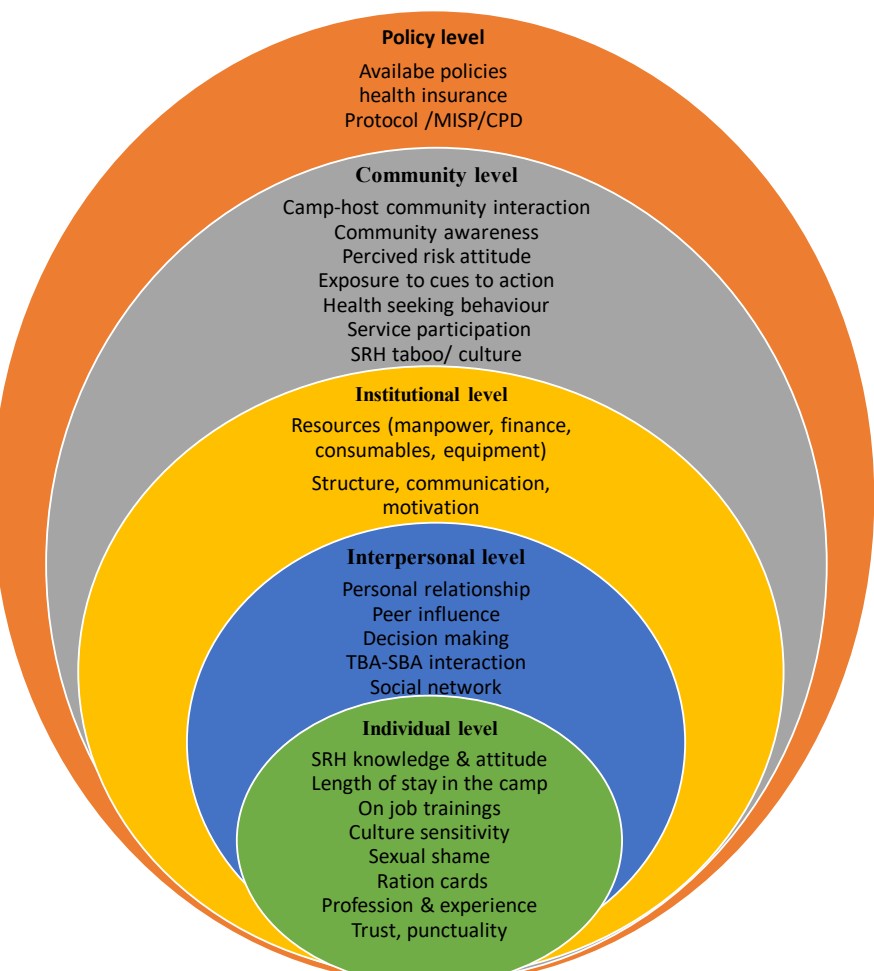

**Figure 1** A socioecological analysis framework of factors that influence HCPs' readiness to provide SRH services to Eritrean refugee women in Asayta refugee camp, Ethiopia (adapted). HCP, healthcare providers; SRH, sexual and reproductive health.

pregnancy, and childbirth. They also want to know how maternity issues might affect their asylum claim. KII aged 20–30 years

Another HCP elaborated her critical observation as follows:

Awareness among refugee women is limited. There is a need for 'repeated SRH counseling sessions' to build trust in SRH care. I believe that women use SRH services not because of better health behaviors, but because of the food supply and the camp's refusal to register children born outside the camp clinic. KII aged 30–40 years

### HCPs are reluctant to provide safe abortion services

Eritrean refugee women in need of abortion often turn to private clinics for abortion services. The HCP at the hospital said:

Oh yes, we have to recognize Ethiopian abortion law. So what's happening on the ground seems to be

less official policy and more 'cultural/religious resistance' to safe abortion services. KII aged 20–30 years

Safe abortion services are generally believed to be not available due to the reluctance of HCPs. While most private clinics offer clandestine abortion service, the cost is prohibitive for refugee women, and the skills of the professionals in private clinics are not reassuring. As a result, refugee women often opt for illegal, self-induced abortions which are likely to be unsafe and put women at risk of severe bleeding. HCP interviewed reported that refugee women with complications following unsafe abortions are admitted to public hospitals.

I have never seen a refugee woman come here (to the refugee clinic) to have an abortion. They do it themselves, self-induced. They buy abortion pills from unknown sources. It is also common to see schoolgirls from the refugee camp buying abortion pills to end a pregnancy. These drugs are purchased without a doctor's prescription. They even have their traditional methods, like using certain traditional medicines,

and they come to us with heavy bleeding. So by the time they come to us, the pregnancy has already been terminated. So we had to refer them to Asayta Hospital, where they receive better treatment like D&C [dilation and curettage]. Curettage removes anything that is left behind. KII aged 20–30 years

### Women's autonomy

Autonomy refers to a woman's power to make decisions about her SRH services free from coercion and violence. Women's self-decision-making power enables individuals, especially marginalised or disadvantaged groups, to have more control over their own lives. For refugee women, autonomy can involve a range of activities aimed at increasing their voice and decision-making power.

Refugee women who arrived at the camp reported having very limited autonomy in sexual and reproductive health issues. I think this is due to limited literacy skills, poor understanding of health issues, inability of HCP to speak the local language, and limited social networks, usually competing with competing childcare responsibilities in large families with six to eight children in a household. KII aged 30–40 years

Women's autonomy could be strengthened through economic empowerment. However, as a KI participant explained below, it is less likely to be pragmatic, given the environmental and economic landscape of the region in favour of refugees.

It is unlikely that the refugees will be able to find a job here. They usually go far to collect firewood in the grassland, making cultural items such as mats and grass rugs, which are used for prayers, raising goats for sale, or working as wage laborers in the construction sector, and factories. They usually work hard to earn little money to buy grains of wheat to feed their children. KII aged 30–40 years

### Women's empowerment

The empowerment of women is crucial for unlocking their full potential. Within refugee camps, empowerment is not only a moral imperative but also vital for the overall well-being and stability of the camps. Empowered women can more effectively nurture their families, actively engage in community development and pave the way for a brighter future for themselves and their children. Unfortunately, Eritrean refugee women lack opportunities to access programmes promoting women's empowerment.

The refugee women are not interested in applying for social assistance, preferring to find suitable work to meet their family's basic needs. They describe life in the refugee camp as a modern prison, even though the security situation here is much better than in Eritrea. KII aged 30–40 years

### Health-seeking behaviour

The SRH needs of both the refugees and women in the hosting communities are found to have many similarities. Among these are having similar SRH needs related to accessing SRH services, cultural/traditional expectations and engaging in harmful traditional practices. However, refugee women were found to have additional SRH needs related to issues of referral systems, gender-based violence, women's empowerment and limited resources in refugee camps. Even though refugees and local people share common sociocultural standards and behaviours, refugee women have better health-seeking behaviour that differs significantly from women in the host community.

### Interpersonal-level factors

The second level of the SEM perspective examines the relationships that prevent women from using SRH services. The main SRH barriers reported consistent with this SEM level were as follows: language barriers, cultural concerns, interactions with others and peer pressure that may provide social support or create barriers that may influence a woman's perception of healthy behaviours.

### Language and cultural barriers

The first and most common challenge for all of us in here and in the community is the language problem. There is no language school or dictionary to learn the local language (Afaraf). I have tried to learn common words to help the women in their mother tongue, but I am not fluent enough to talk to the women and treat them according to their needs. All the women here want to be treated holistically, in the context of their religious, cultural, and traditional values. KII aged 30–40 years

HCPs expect their clients to be able to understand Amharic. If this is not the case, they are expected to bring translators. Clients, on the other hand, especially newly arrived refugees, may not be aware of this problem. Instead, they expect services to be provided in their native local language (Afaraf) and that a translator will be available if HCPs do not speak the language.

The majority of refugee women in the camp do not speak Amharic, and those with whom we cannot communicate have limited (or no) access to SRH care, even when they come and sit in front of us. The cultural and language barrier prevents refugee women from openly expressing their needs because they are not able to clearly state their needs. That's what we find so unfortunate about the language problem in the clinic. KII aged 20–30 years

A KII at the camp described the frustration with translators and how SRH care for adolescents' needs could be compromised because only one interpreter was available at the refugee clinic. Thus, relatives or family members usually assist with translation. The involvement of relatives as interpreters in healthcare may raise ethical

concerns regarding privacy and confidentiality of the services provided.

> Adolescent refugees are the main users of contraceptives. This is often influenced simply by the presence of a translator who is a member of the refugee community. Adolescents are embarrassed to talk about SRH issues and contraceptives when she (translator) is present. They prefer to call my cell phone for private counseling. KII aged 20–30 years

HCPs are accused of showing insensitivity to the cultural practices and religious beliefs of refugee women and of assuming that the way they treat them meets global scientific standards. This insensitivity is expressed in them saying:

> Unfortunately, we 'male HCP' continue to do what we have trained for: 'frequent touching' and 'naked exams'. However, many refugees assume they can be guaranteed a female HCP to address sensitive SRH needs, especially during childbirth. KII aged 30–40 years.

### Peer pressure

Peer influence can be a powerful factor in shaping the SRH needs of refugee women. This happens when individuals feel pressured to conform to the norms, values or behaviours of the group in order to fit in or be accepted.

> We experience that refugee women are often influenced by their peers. This peer pressure may result in positive or negative effects on SRH decision-making. It can also be helpful to seek support from trusted friends, family members, or health care providers when making health behavior decisions. KII aged 20–30 years

### Institutional-level factors

In the third SEM level on institutional factors, HCP reported several problems impeding SRH care that are attributable to the organisational level. These bottlenecks relate to the physical work environment, duties and responsibilities, incentives, overtime pay, laboratory facilities, medications, supplies, professional development plan, organisational communication and decision-making. Unavailability of HCP and essential services such as a central data clerk, laboratory, and pharmacy, poor quality of services, long wait times and prohibitive transportation costs for life-threatening emergencies were the barriers frequently reported to camp officials.

### High staff turnover

HCP cited several reasons for the high turnover in the Asayta refugee camp. These include the following: difficult working conditions, inadequate shelter, harsh living conditions and traumatic experiences. Other factors that may contribute to high HCP turnover include the following: low job satisfaction, inadequate supervision

and limited opportunities for professional development. The high turnover of HCP was mentioned as one challenge to establishing sustainable healthcare in the camp. Major collapses in care delivery were during the war in Northern Ethiopia where the camp was overwhelmed by a large number of refugees. This high turnover worsens during the pandemic. A study participant stated:

> It is difficult to convince RRS administration to accept new practices. It takes time to convince them, and on top of that, key leaders change frequently. The leader you convinced and who accepted your proposal is soon replaced, and the new leader needs another time to understand your plan, which is the biggest challenge to staying on the job. This is extremely concerning in our professional duties. To address the high turnover among HCP in refugee camps, it may be helpful to address the root causes of this problem and implement strategies that improve working conditions, increase job satisfaction, and provide opportunities for professional development and advancement. This could include providing adequate resources and support for HCP, improving housing and living conditions, and offering CPD opportunities. KII aged 30–40 years

### Fragmented healthcare system

HCPs employed in refugee structures function independently of the regional (district) healthcare system and are structurally responsible for the federal RRS. The refugee clinic lacks essential laboratory equipment and supplies, which can prevent refugee women from accessing SRH services. As a result, the clinic often refers its patients to other hospitals, including Asayta Hospital, Dubti Hospital and other specialised hospital in Addis Ababa. This referral process is lengthy and cumbersome and can be challenging for patients seeking treatment, especially if they have limited transportation or financial resources. The lack of essential supplies affects the ability of health workers to make accurate diagnoses and treat patients and can lead to delays in treatment and negative health outcomes.

> Most of the patients are Eritreans because they have no other options to seek treatment and rely on the referral chains in the camp, which are too long and cumbersome. The lack of basic laboratory materials affects the diagnosis and treatment of clients and can lead to delays in care and negative health outcomes. KII aged 30–40 years

Another HCP provider expressed frustration with the complex challenges of the referral system, emphasising the difficulties of SRH services for women from remote locations:

> The referral system in place in Asayta refugee camp may not be accessible or practical for women living in remote or hard-to-reach areas, such as 'way up' in

the mountains and off the road. In these cases, it is difficult or impossible for ambulances to reach these women, instead, mothers in labor may have to be carried to the hospital by others, which can be physically and emotionally exhausting and increase the risk of complications for both the mother and the baby. KII aged 20–30 years

### Community-level factors

The fourth level of SEM focuses on the broader community and societal factors that may impact the SRH needs of Eritrean refugee women living in camps and host communities. These factors include the cultural, social and physical norms that prevail in camps and host communities, as well as the physical and social environments in which people live and work. SEM also looks at the perceived social value of HCP and the perceived stigma associated with seeking health services, as well as satisfaction with local, state and national government policies. In this study, community-level influences on the SRH needs of Eritrean refugees that emerged included factors within the camp setting, such as informal networks, for example, community leaders, and the availability of community resources and interactions with the host community. All of these factors can impact refugees' ability to access the SRH services they need.

### SRH is a taboo topic

It is not uncommon for certain cultural or religious practices to influence attitudes towards SRH issues. Among groups of Eritrean refugees, these practices are seen as incompatible with cultural or religious beliefs and are considered offensive or inappropriate. This makes it difficult for refugee women to access SRH services or make informed decisions about their health and can lead to negative consequences such as social exclusion or stigmatisation.

HCP explained how SRH issues are very much taboo topics:

> SRH issues are taboo topics that are rarely talked about within the family circle. KII aged 20–30 years

Another HCP added:

> Even the thought of SRH services, contraceptives, sex before marriage, and abortion is considered "harmful to religion and culture" because culture and religion dictates remaining virginal until marriage. Adolescents who have premarital sex are considered guilty and can be excluded from the family and society. KII aged 30–40 years

Community perceptions of SRH services can have a significant impact on the uptake of these services. In Afar communities, there are many negative attitudes or stigmas surrounding SRH services, which can make it difficult to access these services. Women view SRH services as inappropriate or immoral, or they may have cultural or religious beliefs that conflict with accessing these services.

This can lead to a lack of support for SRH services within the community and make it difficult for people to access these services even when they are available. On the other hand, in communities where SRH services are viewed positively and supported by community leaders and other influential figures, these services will likely be more widely available and more widely used. HCP we spoke explained that refugee women and especially young adolescents have difficulty accessing SRH services for a variety of reasons, including cultural norms and community perceptions. In Afar cultures, discussions of SRH issues are taboo or stigmatised, which can make it difficult for young people to access information or services. These negative perceptions of SRH services or that the community views that they are inappropriate have created barriers to accessing services. These cultural and community factors can make it difficult for adolescents to access the SRH services they need, which can negatively impact their health and wellbeing. A HCP pointed out:

> When a school girl from the camp comes to the SRH clinic, people think, 'Oh, this girl has already started having sex', and so young people are ashamed to come and ask; they prefer to solve these problems themselves. KII aged 20–30 years

### Insensitivity to cultural practices

Ethiopian health workers in the camp were generally positive about the Eritrean refugee women they care for, emphasising that they strive to provide the same level of care to all women in need. In the Asayta refugee camp, where women come from different cultural backgrounds, they expect these cultural practices to be respected in all circumstances, but the structure of the health system makes this extremely difficult.

> This (sensitivity to cultural practices) is especially important for HCP, who must be sensitive to the cultural practices and beliefs of the refugee women they serve to provide respectful and appropriate care. Majority of HCP are strangers usually have difficulties in responding to the cultural needs of Eritrean refugee women, which can lead to misunderstanding or miscommunication and make it difficult for patients to receive the care they need. In some cases, the cultural insensitivity of health professionals may also contribute to a lack of trust in the healthcare system, which can further impede access to care. KII aged 30–40 years

### Lack of husband involvement in SRH services

The lack of involvement of husbands in SRH services is a significant barrier to the uptake of these services, particularly in areas such as Afar where SRH is seen as a woman's issue or where there are cultural or social norms that discourage men from becoming involved in these issues. Men's lack of participation can negatively impact men's and women's health and well-being by limiting

access to information and services that are important for preventing unintended pregnancies and STI.

> There are several sociocultural barriers such as restrictive norms and stigma around adolescent and young adult sexuality, harmful gender norms, and discrimination by communities, families, partners, and providers. Men have low knowledge about SRH issues. As a result, they do not support their wives to access SRH services. In general, the culture does not support the use of some SRH services, particularly family planning services. As a result, the community does not have positive attitudes toward family planning, and men and elders do not want their wives and children to limit the number of offspring they bear. KII aged 20–30 years

### Gender-based violence

In a humanitarian setting, violence against women is more common with limited access to support systems and basic services. Common forms of violence against women in the Asayta refugee camp include physical, sexual and emotional abuse, as well as financial exploitation and female genital mutilation. These types of violence are perpetrated by individuals, such as intimate partners or family members, or by groups or institutions. Violence against women is normalised and occurs more frequently because of conservative cultural, social and religious norms that condone violence against women and girls as normal, acceptable behaviour. A HCP stated:

> As a concerned professional, I know that violence against women is prevalent in our community and is not effectively addressed due to cultural, social, and religious norms. Violence against women and girls is a normal practice here and is never condemned by the community. Moreover, violence by husbands against their wives tends to be justified. Therefore, it is more difficult to recognize and respond to, as it may be considered a normal or acceptable part of daily life. This can make it more difficult to prevent GBV and provide support and services to those who have experienced it. It is important that communities and societies actively address and work to prevent the normalization of violence against women to create a safer and more equitable environment for all. This can include education and awareness-raising efforts, but also trying to change harmful cultural norms and practices that contribute to GBV. KII aged 30–40 years

### Policy-level factors

This fifth level of the SEM framework adds a macro perspective that considers the broader context. The national, local and humanitarian policies and laws that regulate or support the SRH needs of refugee women are considered.

The 'Out of Camp' policy introduced by the Ethiopian government in 2010 aims to promote the integration of Eritrean refugees into the host society. This includes access to education for refugee students at all levels, including primary, secondary and higher education institutions. The Ethiopian government has taken steps to provide free SRH services in all public health facilities. This is an effective way to encourage women to use these services and improve SRH. However, HCP pointed out that it is also important to address other confounding barriers that prevent women from accessing these services, such as the cost of transportation and housing.

> There are still indirect costs that can place a significant burden on pastoral women, especially those living in humanitarian crises, such as refugees. It might be helpful to think about implementing additional policies or programs to address these types of barriers, such as providing transportation vouchers or subsidies for housing costs. This could help ensure that all people have equal access to the health services they need. KII age 30–40 years

### Weak coordination and collaboration

Lack of coordination and collaboration between different actors of RRS and the Ethiopian Ministry of Health structure and beyond led to inefficiencies and gaps in care. Coordination between the different actors responsible for the refugees is cited as a major challenge in implementing an integrated health system, especially in emergencies.

> Poor coordination among stakeholders has serious consequences. It can lead to duplication, confusion, and misunderstanding, which can ultimately be detrimental to patient health and well-being. It may be helpful to establish clear protocols for better communication and collaboration. This could include regular meetings or a centralized coordination system for sharing information and resources. KII aged 30–40 years

### Training needs of HCP at the refugee clinic

HCPs in the Asayta refugee clinic need specialised training including psychological training to meet the changing health needs of refugee women. They recognised that the SRH needs of refugee women are 'more complex' and 'completely different', requiring intensive preparation in culturally sensitive skills. Despite this realisation, most had not participated in any training that specifically addressed refugee women's healthcare.

> Almost all refugee women have experienced trauma and all demand special needs. HCP need to be trained to recognize these needs and address them in a sensitive and culturally appropriate way. This is a very specific topic area (intercultural competence) that is not even addressed in universities. KII aged 20–30 years

### Health system resilience

Another very important issue that needs to be addressed at the macro level is the resilience of health systems in the region, including in refugee camps, which are not well prepared for natural and man-made emergencies and shocks. As a result, populations lack access to basic health services during times of war, pandemics and natural disasters. The existing health system was easily overwhelmed during the war in northern Ethiopia, the COVID-19 pandemic and the locust invasion, leaving the population without access to essential health services.

> The security situation in the region and even in Ethiopia is volatile; peaceful occasions can rapidly turn into conflicts and devastating situations, as we experienced during the war in northern Ethiopia, when large numbers of people were displaced from northern Afar and Tigray to the town of Asayta, overwhelming the existing health system. At a certain stage of the peace agreement, these IDP are forced to return to their original place of residence. In such cases, the health system is unable to meet the demands of the community, and resilience is severely compromised. Resilience-building strategies should be implemented, including strengthening emergency preparedness and response plans, increasing investment in the health system, and developing a more robust disaster forecasting and analysis system. In addition, partnerships with international organizations and other actors may be beneficial to provide additional resources and expertise to improve the health system in the region. KII aged 30–40 years.

## DISCUSSION

Refugee women encounter significant inequalities in accessing and uptake of SRH services despite substantial influx of migrants towards developing countries. Eritrean refugee women experienced worse health outcomes than the host population. This study identified numerous barriers hindering refugee women's access to appropriate SRH services. Analysed through a SEM framework, these challenges spanned five levels: individual, social, community, institutional and policy. A similar multilevel intervention approach is suggested by Vlahov et al to address the needs of women in humanitarian settings.[58] This reinforces the notion that providing effective SRH services requires collaboration and coordination across various complex systems, as highlighted by studies conducted in Bangladesh and Australia.[7 35]

Individual-level SEM factors such as migration history, education, SRH knowledge, age, culture, socioeconomic background, experience of healthcare and length of stay in the camp were found to shape women's perceptions of their health and well-being, as well as their access to and utilisation of SRH services. These results are in line with studies conducted[39 59] and Multicultural Centre for Women's Health,[30] which showed poor knowledge about

SRH. Interactions with people, and peer pressure, which can provide social support or create barriers might affect a woman's healthy behaviour perceptions. Peers, partners and clan members influence SRH service use behaviours. Interaction between HCPs at the same time influences health service uptake.[60 61] Language and cultural expectations were prevalent and predominant themes that emerged as common SRH barriers among refugee and host community women in Asayta. Differences in language expectations were reported to affect SRH care for both HCP and women. Similar study findings highlight that different expectations between HCP and women are significantly affecting service quality.[36 37 61–63] High staff turnover, lack of essential services, laboratory, pharmacy, long waiting times, frequent referral and transport problems were the most common barriers reported at the health facility level by study participants. Similar findings are reported from refugee camps in Kenya.[64 65]

Even though GBV is a widespread problem affecting people around the world and it is particularly prevalent in conflict and displacement situations such as refugee camps and temporary settlements, it is concerning to witness the normalisation of GBV. During migration and human trafficking, there are horrific stories of violence and yet are still taken as normal in the system. This finding is supported by a study conducted in Nigeria and many other countries where community acceptance of this abusive relationship is more entrenched and where women suffer in silence as they are often protected by family secrecy, cultural norms, fear, shame and associated social stigma and therefore remain silent.[66–70] This is mainly due to the general religious teachings that women must submit and obey their partner as the head of the family.[69 71] SRH topics in the camp are also taboo topics similar to other studies.[72] On a similar token, HCPs in Asayta generally do not perform safe abortions due to prevailing negative perceptions. Women who need abortions often turn to private health facilities. The same findings are reported in many African countries where SRH is largely a taboo topic, intentionally avoided or not addressed either by the HCP or the healthcare seeker because it is socioculturally unacceptable or connected with strong feelings of shame.[6 73 74] A macro perspective model that considers the broader context focus was on national, local and humanitarian policies and laws that regulate or support the SRH needs of Eritrean refugee women in the Asayta camp. These included policies on community integration, livelihood and access to services such as healthcare, education, housing, epidemics control and disease management. In agreement with other studies,[39 75] HCP in this study indicated that factors at all levels of SEM influence the SRH needs of Eritrean refugee women. HCP perceived that Eritrean refugee women SRH needs and engagement with care varied across SEM perspectives.

The SRH needs of refugees and host communities have many similarities. The mentioned difference in a health-seeking difference could be due to the mandatory linkage

of humanitarian aid services with the use of maternal health services in the camp clinic. This can be an effective strategy to promote SRH-seeking behaviour, but it is also important to ensure that these services are culturally appropriate and meet the needs of the population they are intended to serve.

Under the available laws, refugees in Ethiopia are entitled to protection from persecution and to the same rights and privileges as Ethiopian citizens, with some exceptions. These include the right to education, the right to work and engage in economic activity and the right to freedom of movement within certain limits.[76 77] HCP reiterated the importance of addressing indirect cost barriers that can prevent women from accessing these services, such as transportation and housing costs. Although Eritrean refugees have a right to access health services, including SRH care, they face multiple complexities where access to services is limited by several factors, including geographic remoteness, lack of resources and sociocultural barriers.

The lack of alignment between the national health system and the RRS system of refugee creates barriers for HCPs in those camps, and they face challenges in accessing continuous professional development opportunities. As a result, most camp providers lack access to the latest advancements in disease management, treatment protocols and other essential areas, potentially compromising the quality of care they can deliver.

Another crucial issue is the resilience of the health system in the refugee setting. This situation is not well prepared for emergencies and shocks such as disease outbreaks, natural disasters or conflict. The existing health system was easily overwhelmed by triple disasters (war in Northern Ethiopia, COVID-19 pandemic and locust invasion), leaving the population without access to basic health services. Eritrean refugees were among the most vulnerable populations affected by the grave civilian atrocities, displacement and deteriorating humanitarian situation in northern Ethiopia.[78] Thus, effective refugee policies that are designed to protect the rights and well-being of refugees, while also taking into account the needs and concerns of host communities, are needed more than ever. This also highlights the need for increased support from the international community to help Ethiopia meet the needs of refugees and improve the resilience of its health system.

### Strengths and limitations

The SEM enabled this study to comprehensively explore the multilevel factors influencing refugee women's SRH needs, from individual experiences to societal structures, revealing insights not captured by previous studies in the country. This study is the first attempt to examine SRH service utilisation from the perspective of KIR approach. This again provides a unique and insightful starting point for understanding the factors that influence SRH service utilisation among refugee women in a humanitarian crisis settings. Exploring SRH studies presents unique challenges due to the sensitive nature of the topic. Participants

may hold biases or fears regarding the researchers' intentions, potentially introducing bias into the study. Additionally, building trust and rapport with participants can be difficult, impacting the study's success. HCPs were recruited from the Asayta district; therefore, comparing results with other sites may be difficult because different policies and programmes apply in different refugee camps in different regions. To promote reflexivity and mitigate these biases, triangulation of evidences through focused observation was done.

### Implications for research, policy and practice

This qualitative study provides invaluable insight into the SRH of Eritrean refugee women from the perspectives of HCP. The study findings confirmed that SRH topics are taboos and act as a barrier to access SRH services. Likewise, it should be ensured that HCP working in the area of SRH should be equipped with the proper training on cultural, compassionate and respectful competency care approaches. SRH consultations should take place in a safe and supportive manner. Healthcare facilities in the camp should also create private spaces for consultations to maintain the confidentiality of SRH services. Further research is needed to explore the knowledge and attitudes of HCP in SRH care approaches and in understanding the barriers and facilitators towards providing comprehensive SRH service. This has the potential to facilitate the proper use of SRH services consequently leading to overall improvements to women health.

Even though SRH is included in Ethiopian social policy, the implementation of rules and regulation are not available or very minimal. It is critical to establish national policies and guidelines describing the standard of care in the area of SRH and clarify providers' roles in offering SRH advice. The findings may also be useful for public health implementers and policy evaluators, as they could broaden their understanding of what works more effectively and for whom, in fragile humanitarian contexts, with the aim of improving SRH service use among refugee women.

### CONCLUSIONS

Humanitarian care settings are often characterised by the breakdown of basic health services. This study highlights the higher unmet SRH needs of Eritrean refugee women in Ethiopia, with severely limited service availability. The findings showed the need to develop interventions that simultaneously target multiple levels of SEM highlighting the importance of evidence-based solutions at the individual, group, community, institutional and policy levels to address the SRH needs of Eritrean refugee women. For example, at the individual level, women's past experiences can pose challenges to SRH service use. At the interpersonal level, lack of cultural competence, communication difficulties, cultural differences, misunderstandings and misconceptions were found to impact service delivery. The results have the potential to have an even greater

impact, as lessons learnt from SRH decision-making can be applied to other general health services. Implementation of lessons learnt about refugee health in Ethiopia should also take into account differences in culture and the traditional needs of the target populations. Underlying all solutions and their implementation should be the empowerment of women to optimise their knowledge of SRH.

Given the COVID-19 pandemic and fragile situation, health needs could be challenged at any time, and preparation is crucial at all levels of the SEM to ensure health system resilience. Future public health interventions must build on stakeholder input and address all aspects of SEM to achieve the SDG and reduce maternal mortality. This could involve advocating for increased resources and funding, establishing partnerships for collaboration and distributing supplies.

The need for better collaboration between the RRS health system and the regional (district) health system is very paramount. Developing a comprehensive and practical programme, as well as legislative and policy support for this issue, can provide the foundation for SRH care for this group of women in refugee camps. Training HCP from the local area who are familiar with culture and language can provide the services effectively. The lessons learnt through HCP are important for both policy and practice, as they provide a framework for planning, developing and implementing interventions for governments and partners charged with improving the use of SRH services. It is also important to work with community leaders and other influential figures to promote gender equality and address harmful cultural norms.

**Acknowledgements** We are very grateful to all study participants for their time and valuable contributions. Our appreciation goes to the entire NIIDS project team in Switzerland and Ethiopia. We are also grateful to Karen Maigetter for her invaluable assistance with the language edits.

**Contributors** NBZ contributed to the conceptualisation, methodology, investigation, analysis, and writing of the original draft. SM, AAM, DHP and NP-H provided critical review and supervision. CA, AC and RT contributed to project development and coordination. SM, AAM, RT, CA, AC, DHP and NP-H contributed to and reviewed the article prior to submission. All authors are a guarantors, have read and approved the final version.

**Funding** The 'HEALTH NIIDS' project supported this work through the generous financial support of the Stanley Thomas Johnson Foundation and the State Secretariat for Education, Research and Innovation, Switzerland with grant number 1053-KF. The funders have no role in the design, analysis and dissemination of the findings.

**Competing interests** The authors declare that they have no competing interests

**Patient and public involvement** Patients and/or the public were not involved in the design, or conduct, or reporting, or dissemination plans of this research.

**Patient consent for publication** Consent obtained directly from patient(s).

**Ethics approval** This study involves human participants and was approved by the Ethics Committee of Northwestern and Central Switzerland (EKNZ; EKNZ No. 2020-02154) in Basel, Switzerland, and the Research Ethics Committee of the Armauer-Hansen Research Institute in Ethiopia as part of the 'HEALTH- NIIDS' project. Participants gave informed consent to participate in the study before taking part.

**Provenance and peer review** Not commissioned; externally peer reviewed.

**Data availability statement** All data relevant to the study are included in the article or uploaded as supplementary information.

**ORCID iDs**
Nejimu Biza Zepro http://orcid.org/0000-0001-8502-7605
Sonja Merten http://orcid.org/0000-0003-4115-106X

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
