## [Reviewer comments · BMJ Open]

ARTICLE DETAILS

TITLE (PROVISIONAL)	Navigating Challenges: A socio-ecological analysis of sexual and reproductive health barriers among Eritrean refugee women in Ethiopia, using a key informant approach
AUTHORS	Zepro, Nejimu; Medhanyie, Araya; Probst-Hensch, Nicole; Chernet, Afona; Tschopp, Rea; Abongomera, Charles; Paris, Daniel; Merten, Sonja

VERSION 1 – REVIEW

REVIEWER	Sifat, Ridwan Islam University of Maryland Baltimore County, School of Public Policy
REVIEW RETURNED	04-Jan-2024

GENERAL COMMENTS	The existing information regarding the previous study's findings does not offer readers insights into the foundation and methodology of the present study. Nevertheless, to provide a more comprehensive context, it is advisable to incorporate perspectives from diverse countries in the introduction and background sections. Additionally, the methodology section requires further elucidation regarding the rationale behind the selection of specific methods, techniques, and sampling procedures. Furthermore, the results paragraph needs to be summarized in order to provide a more fluent section for the readers. Finally, what is the take-home message of this manuscript? What are the exact recommendations? You may write some potential recommendations. You need to rearrange the conclusion part for more analysis. The authors should explore additional perspectives from minority populations in different countries to enrich the study's findings and implications. You may cite this article: https://doi.org/10.1007/s13178-022-00758-z
--

REVIEWER	Huda, Md. Nazmul Independent University
REVIEW RETURNED	22-Jan-2024

GENERAL COMMENTS	Abstract: 1. Good topic. However, this sentence ('there is a need') seems unclear to readers. What does 'need' indicate here? Also, what challenges did you indicate by the phrase 'these challenges'? Introduction Paragraph 1: 1. Migration health or migrants' health? Good story with a focus on SDGs. Paragraph 2 & 3:
--

	2. Avoid writing small paragraphs without a story. These paragraph's story is relatively weak without a meaningful story. Rewrite them or merge with a good story. Paragraph 6: 3. This paragraph attempted to define SRH. However, the clear-cut definition is still missing. Please provide a concrete definition. Paragraph 7: 4. I think you can merge it with the above paragraph. Paragraph 8: 5. The use of SEM model is good. However, I think the descriptions of challenges at different levels require more review of the literature, leading to gaps in the existing literature and your aim. Final paragraph: 6. Check font size and type. 7. What is the difference between themes and categories? Do you need to mention both? Materials and methods: 1. Please cite some literature which you reviewed to develop your interview guide. 2. I am not sure whether you included field notes in the analysis. Please clarify it. Findings: 1. The first six lines can be part of the method section. Not sure why here. 2. I think it is better to provide operational definitions of all five levels of SEM. Otherwise, it is confusing for the readers. Because some factors can be overlapping at different levels. A definition of each level can help categorise them and describe your findings. 3. You are indenting and applying quotation marks simultaneously. Do you think you need both? Please check these in line with the referencing style you used in this manuscript. Also, if the quotes are less than 40 words, use quotation marks and merge with the paragraph about your analysis. If there are more than 40 words, indent only. Do not use italic words. Check the journal's requirements and apply the same for all quotes in this paper. 4. Avoid putting two quotes together as this prevents in-depth analysis and detract readers' attention. Rather, mention one quote with analysis followed by another. Discussion 1. Rewrite the first paragraph and describe your findings briefly. You can also describe your contribution to the scholarship. 2. This paragraph ('due to structural...' looks orphan to me. Please rewrite or merge with existing paragraphs coherently. 3. Describe strengths first followed by the limitations. Conclusion 1. Write a separate section on the policy implications, followed by the conclusions. Overall: It is an interesting paper. Please address the comments and check grammar and typos in all sections of the manuscript.
--	---

VERSION 1 – AUTHOR RESPONSE

Authors' response:

Thank you for your constructive feedback on our manuscript. We appreciate your time, insights and suggestions for improving the clarity and comprehensiveness of the study. Below is the outline of the steps we took to improve each sections of the article to address your relevant suggestions:

Introduction: it is indeed important to provide a more comprehensive overview of the situation starting from global diverse country perspectives. We revised the introduction section to incorporate perspectives from diverse countries, providing a broader context. We added the few more papers from diverse context including the article you suggested about the SRH situation of Rohingya refugee women in Bangladesh

Methodology: We understand the importance of providing readers with a clear understanding of the methodology of our study and the rationale behind. To address this, we tried to elucidate the methodology section for better understandings of specific methods, techniques, and sampling procedures we used.

Results: We rewrote some of the paragraphs in the result section to provide a better, fluent, and cohesive follow of ideas.

Conclusion: Thank you for the feedback. Likewise, to modify the paragraph to reorganize and articulate the key findings and their implications more effectively, this section is also rewritten

Recommendations: we tried to include more specific recommendations, such as exploring a more comprehensive interventions that can address the five levels of SEM simultaneously.

We are committed to making these revisions to ensure that our manuscript meets the highest standards of clarity and comprehensiveness. We appreciate your guidance in this process and look forward to hear any additional comments you might have in the revised manuscript. Thanks you

Reviewer: 2

Dr. Md. Nazmul Huda, Independent University, Western Sydney University - Campbelltown Campus

Comments to the Author:

Abstract:

1. Good topic. However, this sentence ('there is a need') seems unclear to readers. What does 'need' indicate here? Also, what challenges did you indicate by the phrase 'these challenges'?

Thank you for raising this important point. We agree that the original sentence lacked clarity. We rephrased the title now as "Navigating Challenges: A socio-ecological analysis of sexual and reproductive health barriers among Eritrean refugee women in Ethiopia, using a key informant approach". We feel that this title reflects the research question, study design and study subjects. This was also a comment from the journal editor. We were highlighting the need for culturally appropriate and effective interventions to address the numerous challenges Eritrean refugee women face in accessing essential SRH services in Asayta camp. The term "these challenges" was to mean limited awareness and knowledge, cultural taboos, lack of safe spaces, inadequate healthcare facilities, and restrictive policies.

Introduction

Paragraph 1:

1. Migration health or migrants' health? Good story with a focus on SDGs.

Thank you for this notice. As it is the first paragraph, we aimed to convey the broader idea of the field of study encompassing various health aspects related to migration including the health of migrants, the impact of migration on health systems, and the health risks associated with migration processes. of course, both "migration health" and "migrant health" are terminologies used interchangeably depending on the specific context related to the health of individuals who migrate.

Paragraph 2 & 3:

2. Avoid writing small paragraphs without a story. These paragraph's story is relatively weak without a meaningful story. Rewrite them or merge with a good story.

Thank you for the feedback. We have rewritten this statements by merging paragraph 2 and 3 with more stories from the previous scientific studies conducted to elaborate the SRH needs of refugee women in diverse context. In addition, a literature also recommended by another reviewer is added.

Paragraph 6:

3. This paragraph attempted to define SRH. However, the clear-cut definition is still missing. Please provide a concrete definition.

Thank you for this comment. We have rewritten this statement. Sexual and Reproductive Health, doesn't have a single, universally agreed-upon "concrete definition" because of the concept encompasses various aspects of health across multiple aspects of health related to sexuality and reproduction throughout people's lives. However, we have put a definition by UNFPA and ICPD. Hopefully, this clarifies the intended definition of SRH while acknowledging its multifaceted nature.

Paragraph 7:

4. I think you can merge it with the above paragraph.

Thanks you. Recommendation well-noted and the paragraph having similar message are now merged

Paragraph 8:

5. The use of SEM model is good. However, I think the descriptions of challenges at different levels require more review of the literature, leading to gaps in the existing literature and your aim.

Thank you for your feedback. We appreciate your recognition of the SEM model's value in the paper. You're absolutely right that a more comprehensive review of the literature can strengthen the descriptions of challenges at different levels. We revised this paragraph in order to strengthen the literature review despite word count limitations to put exhaustive comprehensive review findings across each SEM levels.

Final paragraph:

6. Check font size and type.

Thank you for the comment. Now this is adjusted to make the entire paragraph uniform

7. What is the difference between themes and categories? Do you need to mention both?

Thank you for this question. In most of qualitative studies researchers use them interchangeably but categories are the initial labels used to organize and code raw data into more descriptive discrete elements, while themes are the broader overarching patterns or concepts that emerge from the analysis of coded data, providing deeper insight into the underlying meaning of the data. They usually represent the underlying meanings, ideas, or narratives from individual categories or codes. Themes are more interpretive and abstract than categories, capturing the deeper significance or implications of the data.

Materials and methods:

1. Please cite some literature which you reviewed to develop your interview guide.

Thank you for this comment. As suggested, relevant literatures are now cited in the revised manuscript line numbers 214-215.

2. I am not sure whether you included field notes in the analysis. Please clarify it.

Yes. We incorporated field notes into our analysis process. We used field notes to inform coding and understand the context better. Additionally, we used field notes through focused observation as a source of triangulation to enhance the credibility and trustworthiness of findings. This statement is included in methods section (line 228-229) of the revised manuscript.

Findings:

1. The first six lines can be part of the method section. Not sure why here.

Thank you. Authors agree with your suggestion and these specific line are relocated to the method section.

2. I think it is better to provide operational definitions of all five levels of SEM. Otherwise, it is confusing for the readers. Because some factors can be overlapping at different levels. A definition of each level can help categorise them and describe your findings.

Thank you for this comment. We rephrased this section. We considered your suggestion and tried to add a sentence at each levels of SEM to enhance clarity and help readers better understand how factors are categorized and interact across different levels.

3. You are indenting and applying quotation marks simultaneously. Do you think you need both? Please check these in line with the referencing style you used in this manuscript. Also, if the quotes are less than 40 words, use quotation marks and merge with the paragraph about your analysis. If there are more than 40 words, indent only. Do not use italic words. Check the journal's requirements and apply the same for all quotes in this paper.

Thank you for this relevant point. We checked the journal requirements and we modified the writing of the result section throughout the paper as per your suggestion.

4. Avoid putting two quotes together as this prevents in-depth analysis and detract readers' attention. Rather, mention one quote with analysis followed by another.

Thank you for this notice. This is now modified in the entire manuscript.

Discussion

1. Rewrite the first paragraph and describe your findings briefly. You can also describe your contribution to the scholarship.

Thanks for this point. We have now modified the first paragraph of the discussion section.

2. This paragraph ('due to structural...' looks orphan to me. Please rewrite or merge with existing paragraphs coherently.

Thanks for this point. We have now modified the entire paragraph. We rewrote this section to make it clearer for the reader. It reads now: The lack of alignment between the national health system and the RRS system for refugees creates barriers for HCPs in those camps face challenges in accessing continuous professional development (CPD) opportunities. As a result, most camp providers lack access to the latest advancements in disease management, treatment protocols, and other essential areas, potentially compromising the quality of care they can deliver

3. Describe strengths first followed by the limitations.

Certainly. We have now modified the entire section on strength and limitations

Conclusion

1. Write a separate section on the policy implications, followed by the conclusions.

Thanks for this point. We have now added separate section about research, practice and policy implications right before the conclusion section.

Overall: It is an interesting paper. Please address the comments and check grammar and typos in all sections of the manuscript.

We are glad to hear this. We have carefully reviewed your comments and appreciate your suggestions. We are happy to address any further points (if any). Thanks a lot. Your feedback indeed improved our paper.

VERSION 2 – REVIEW

REVIEWER	Sifat, Ridwan Islam University of Maryland Baltimore County, School of Public Policy
REVIEW RETURNED	15-Mar-2024
GENERAL COMMENTS	Thank you for revising the manuscript.